# Bioinformatics Analysis of the Prognostic Significance of CAND1 in ERα-Positive Breast Cancer

**DOI:** 10.3390/diagnostics12102327

**Published:** 2022-09-27

**Authors:** Rashed Alhammad

**Affiliations:** Department of Pharmacology, Faculty of Medicine, Kuwait University, Kuwait City 13110, Kuwait; rashed.alhammad@ku.edu.kw

**Keywords:** CAND1, Erα, breast cancer, bioinformatics, metastasis, survival, prognosis

## Abstract

The identification of novel prognostic biomarkers for breast cancer is an unmet clinical need. Cullin-associated and neddylation-dissociated 1 (CAND1) has been implicated in mediating carcinogenesis in prostate and lung cancers. In addition, CAND1 is an established prognostic biomarker for worse prognosis in liver cancer. However, the prognostic significance of *CAND1* in breast cancer has not yet been explored. In this study, Breast Cancer Gene-Expression Miner (Bc-GenExMiner) and TIMER2.0 were utilized to explore the mRNA expression of *CAND1* in ERα-positive breast cancer patients. The Kaplan–Meier plotter was used to explore the relationship between *CAND1* expression and several prognostic indicators. The Gene Set Cancer Analysis (GSCA) web server was then used to explore the pathways of the genes that correlate with *CAND1* in ERα-positive breast cancer. Immune infiltration was investigated using Bc-GenExMiner. Our bioinformatics analysis illustrates that breast cancer patients have higher *CAND1* compared to normal breast tissue and that ERα-positive breast cancer patients with a high expression of *CAND1* have poor overall survival (OS), distant metastasis-free survival (DMFS), and relapse-free survival (RFS) outcomes. Higher *CAND1* expression was observed in histologic grade 3 compared to grades 2 and 1. Our results revealed that *CAND1* positively correlates with lymph nodes and negatively correlates with the infiltration of immune cells, which is in agreement with published reports. Our findings suggest that *CAND1* might mediate invasion and metastasis in ERα-positive breast cancer, possibly through the activation of estrogen and androgen signaling pathways; however, experiments should be carried out to further explore the role of *CAND1* in activating the androgen and estrogen signaling pathways. In conclusion, the results suggest that *CAND1* could be used as a potential novel biomarker for worse prognosis in ERα-positive breast cancer.

## 1. Introduction

*CAND1* gene encodes the CAND1 protein, which is predominantly localized in the cytoplasm [1]. CAND1 binds to unneddylated cullins [1,2,3] and regulates cullin–RING ubiquitin ligases [4,5,6]. These ligases play a role in the ubiquitinoylation of proteins that are degraded by the proteasome system and control the stability of substrates involved in transcription and cell cycle [1,7,8,9]. In cancer, these ligases are frequently dysregulated [7,10]. It has been shown that several members of the ubiquitin-ligase machinery are promising molecular targets for breast cancer therapeutic, including OTUB1 [11,12].

CAND1 has been implicated in mediating carcinogenesis, in which the downregulation of CAND1 induces apoptosis in prostate cancer cell lines [13]. Other published reports show that CAND1 is overexpressed in prostate cancer tissue compared to normal tissue [14,15]. It has been proposed that high CAND1 expression increases the degradation of tumor suppressor proteins, thereby inhibiting apoptosis [15]. In addition, CAND1 overexpression has been shown to promote the proliferation of prostate cancer by stabilizing PLK4 [14]. CAND1 has also been implicated in mediating proliferation and metastasis in lung cancer, in which targeting CAND1 by miR-33a decreases the proliferation and migration [16]. Moreover, it has been shown that *CAND1* correlates with poor OS in liver cancer, in which silencing *CAND1* suppresses the proliferation of liver cancer by inducing caspase-8/RIP1-dependent apoptosis [17]. *CAND1* overexpression has also been observed in oligodendroglial cancer compared to normal tissue [18]. Additionally, targeting CAND1 with miR-148b-3p regulates Schwann cells migration [19]. miR-148b-3p has also been implicated in mediating the proliferation and migration of breast cancer cell lines [20,21].

Taking into account the fact that *CAND1* overexpression has been shown to promote cancer proliferation and metastasis in several cancers, including lung, prostate, and liver cancers [11,12,13,14,15], and that CAND1 interacts with miR-148b-3p, which is a mediator of ERα-positive breast cancer cell metastasis [17,18,19], it is crucial to investigate the prognostic value of *CAND1* in ERα-positive breast cancer. Moreover, given that there is a great demand for new biomarkers to predict outcomes and enhance individualized treatment due to breast cancer heterogeneity and that there is a lack of information about *CAND1*′s potential as a breast cancer prognostic biomarker, the aim of this study was to explore the significance of *CAND1* in ERα-positive breast cancer by utilizing several bioinformatics tools.

Prompted by the lack of information about *CAND1*’s potential as an ERα-positive breast cancer biomarker, comprehensive bioinformatics analysis was carried out utilizing several databases to evaluate the significance of *CAND1* as a diagnostic tool. Therefore, we evaluated the significance of *CAND1* expression in ERα-positive breast cancer via bioinformatics analysis of the clinical indicators and survival data in several large online databases, including Bc-GenExMiner and TIMER2.0. In this research, the Kaplan–Meier plotter was utilized to assess the relationship between *CAND1* expression and several prognostic indicators, whereas the GSCA web server was used to explore the pathways of the genes that correlate with *CAND1.* In addition, the cBio Cancer Genomics Portal was utilized to assess the relationship between *CAND1* expression and different clinical indicators, including lymph nodes and histologic grades. The Bc-GenExMiner web server was also utilized to explore the infiltration of immune cells.

## 2. Materials and Methods

### 2.1. Investigating CAND1 mRNA Expression in Breast Cancer Tissue and Normal Breast Tissue

The mRNA expression of CAND1 in breast cancer and normal breast tissue was investigated using two different databases. Bc-GenExMiner version 4.1, which is a mining tool of published annotated genomic data, was utilized to investigate the expression of CAND1 in normal breast tissue (n = 92), tumor-adjacent tissue (n = 89), ERα-positive breast cancer tissue (n = 530), and ERα-negative breast cancer tissue (http://bcgenex.ico.unicancer.fr/BC-GEM/GEM-Accueil.php?js=1) (accessed on 31 August 2022) [22]. The TIMER2.0 database was also utilized to explore the expression of CAND1 in normal breast tissue (n = 1093) and in breast cancer (n = 112) (http://timer.cistrome.org/) (accessed on 20 August 2022) [23].

### 2.2. Kaplan–Meier Survival Curve Analysis

The Kaplan–Meier plotter website (http://kmplot.com/analysis/) (accessed on 15 August 2022) was utilized to explore the prognostic value of *CAND1* in ERα-positive breast cancer patients [24]. The website was used to generate Kaplan–Meier plots for *CAND1* in ERα-positive breast cancer patients by selecting the following parameters: OS, DMFS, and RFS; probe set option: all probe sets per gene; split patients by: upper quartile; follow-up threshold: 240 months [25]. Logrank *p*-values of <0.05 were considered statistically significant.

### 2.3. The Infiltration of Immune Cells

Pearson’s correlation coefficients between gene markers of immune cells and *CAND1* were investigated using bc-GenExMiner [26]. Several gene markers of immune cells were included in the analysis, such as M1 and M2 macrophages, T cells, natural killer cells, and B cells. A *p*-value of <0.0001 was considered significant.

### 2.4. The Correlation between CAND1 mRNA and Neoplasm Histologic Grades, Hormone Therapy, and Lymph Nodes in ERα-Positive Breast Cancer Patients

The cBio Cancer Genomics Portal (MA, USA) (http://cbioportal.org) (accessed on 18 May 2022) was utilized, in which the METABRIC dataset was downloaded, which contains 1900 breast cancer patients (Appendix A) [27]. The distribution of the expression of CAND1 mRNA, patients receiving hormone therapy, neoplasm histologic grades, and lymph nodes examined positive were measured across samples. CAND1 expression levels were considered high if their z-scores were higher or equal to the 75th percentile of the distribution [28].

### 2.5. Identification of Genes That Correlate with CAND1 in ERα-Positive Breast Cancer Patients

Genes that positively correlate with CAND1 in ERα-positive breast cancer patients with a Pearson’s correlation coefficient of >0.5 and a *p*-value of <0.0001 were identified using bc-GenExMiner (Table 1). Moreover, miRNAs that target CAND1 were identified (Appendix A).

### 2.6. Association of Genes That Correlate with CAND1 in ERα-Positive Breast Cancer with Signaling Pathways

The GSCA web server (Houston, TX, USA) (http://bioinfo.life.hust.edu.cn/GSCA/#/) (accessed on 10 August 2022) was utilized to explore the involvement of the genes that correlate with CAND1 in activating and inhibiting pathways related to cancer, such as RTK, RAS/MAPK, estrogen receptor signaling, androgen receptor signaling, and TSC/mTOR [29]. Breast-invasive carcinoma patients with available expression data and who had paired samples (paired tumor–normal tissue) were included in the analysis.

### 2.7. Statistical Analysis

The grpahs were ploted using PRISM 8 GraphPad Inc. software (San Diego, CA, USA). The Kruskal–Wallis test was used to assess the significance of the differences between more than two groups, while the Mann–Whitney test was used to assess the significance of the differences between two groups.

## 3. Results

### 3.1. CAND1 mRNA Is Upregulated in ERα-Positive Breast Cancer Patients Compared to Tumor-Adjacent and Normal Breast Tissue

*CAND1* mRNA expression in ERα-positive breast cancer patients was explored using Bc-GenExMiner and TIMER2.0. Significant upregulation of *CAND1* mRNA expression was observed in breast cancer tissue compared to normal breast tissue (Figure 1a,b). Significant upregulation of *CAND1* mRNA expression was also observed in ERα-positive breast cancer patients compared to ERα-negative breast cancer, tumor adjacent, and normal breast tissue (Figure 1a).

### 3.2. CAND1 mRNA Expression Correlates with Poor Prognosis in ERα-Positive Breast Cancer Patients

The prognostic value of CAND1 in ERα-positive breast cancer was explored using the Kaplan–Meier plotter website. It was observed that CAND1 significantly correlates with worse OS (Figure 2a), worse DMFS (Figure 2b), and worse RFS (Figure 2c) in ERα-positive breast cancer patients.

### 3.3. CAND1 Correlates Negatively with Various Gene Markers of Immune Cells in ERα-Positive Breast Cancer Patients

Taking into account that *CAND1* correlates with poor prognosis in ERα-positive breast cancer patients and that the infiltration of immune cells correlates with better prognosis in breast cancer [30,31,32], an investigation of the type of the immune cells infiltrated in ERα-positive breast cancer patients was conducted. The results indicated that *CAND1* correlates negatively with various immune cells, including T cells, B cells, monocytes, M1 and M2 macrophages, and neutrophils (Figure 3).

### 3.4. CAND1 mRNA Positively Correlates with Neoplasm Histologic Grades, Hormone Therapy, and Lymph Nodes in ERα-Positive Breast Cancer Patients

The METABRIC dataset in the cBio Cancer Genomics Portal was explored to validate our observation that CAND1 correlates with worse prognosis in ERα-positive breast cancer patients. CAND1 mRNA significantly correlates with lymph nodes examined positive in ERα-positive breast cancer patients (Figure 4a). A significant positive correlation was observed between CAND1 mRNA expression and neoplasm histologic grades in ERα-positive breast cancer patients, in which significant upregulation of CAND1 was observed in grade 3 compared to grades 2 and 1 (Figure 4b). In addition, significantly higher CAND1 mRNA was observed in ERα-positive breast cancer patients who received hormone therapy compared to ERα-positive breast cancer patients who did not receive hormone therapy (Figure 4c).

### 3.5. Identifying Genes That Positively Correlate with CAND1 in ERα-Positive Breast Cancer Patients

It was vital to investigate the genes that show a significant positive correlation with *CAND1* in ERα-positive breast cancer patients to explore the pathways in which *CAND1* exerts its effect on prognosis, as described in the summary of the experimental design (Figure 5).

The genes that showed a significant positive Pearson’s correlation (correlation coefficient > 0.5) with *CAND1* in ERα-positive breast cancer patients were selected (Table 1).

### 3.6. Genes That Correlate with CAND1 in ERα-Positive Breast Cancer Patients Are Involved in the Activation of the Androgen and Estrogen Signaling Pathways

To explore the function of CAND1 in ERα-positive breast cancer, genes that showed a significant positive correlation with CAND1 were selected to identify the potential pathways that they are involved in. It was observed that the genes that significantly correlate with CAND1 expression in ERα-positive breast cancer patients were involved in the activation of the androgen and estrogen signaling pathways (Figure 6). A summary of the main results is shown in Appendix A.

## 4. Discussion

The results showed that *CAND1* mRNA expression was upregulated in breast cancer patients compared to tumor-adjacent and normal breast tissue and that *CAND1* expression was upregulated in ERα-positive breast cancer compared to ERα-negative breast cancer, suggesting that *CAND1* might predict prognosis in ERα-positive breast cancer. A published report agrees with our findings, showing that *CAND1* was upregulated in ERα-positive breast cancer compared to ERα-negative breast cancer [33]. In addition, *CAND1* upregulation compared to normal tissue has also been observed in other cancers, including prostate and liver cancers [15,17].

The prognostic value of *CAND1* in ERα-positive breast cancer was further investigated using the Kaplan–Meier plotter, in which *CAND1* showed to correlate positively with worse prognosis in ERα-positive breast cancer, suggesting that *CAND1* might act as a predictive biomarker for worse prognosis in ERα-positive breast cancer. Although the prognostic value of *CAND1* in ERα-positive breast cancer has not been extensively studied, it has been suggested that *CAND1* correlates with worse OS in breast cancer [34]. Additionally, *CAND1* has been shown to serve as a potential biomarker for worse prognosis in liver and prostate cancers [15,17]. Moreover, the significant upregulation of *CAND1* in neoplasm histologic grade 3 compared to grades 2 and 1 supports our previous observation that *CAND1* correlates with worse prognosis in ERα-positive breast cancer. Interestingly, CAND1 interacts with CUL4, which is a cullin family member that has been shown to correlate with poor OS in breast cancer [35,36,37]. TUBB, which interacts with CAND1 [4], has also been shown to correlate with poor OS in ERα-positive breast cancer [38].

To further validate our observation that *CAND1* correlates with poor prognosis in ERα-positive breast cancer, the infiltration of immune cells was investigated, as infiltration correlates with better prognosis in breast cancer [30,31,32]. The significant negative correlations between *CAND1* and several immune cells agree with our previous observation that *CAND1* correlates with poor prognosis in ERα-positive breast cancer. Although the correlation between *CAND1* and the infiltration of immune cells in breast cancer has not been studied before, a published report showed that the infiltration of CD4 memory T cells and neutrophils correlates positively with *CAND1* in lung adenocarcinoma [39]. This positive correlation might be due to the fact that the authors used a different approach to assess the correlation. The authors investigated the correlation between the infiltration of immune cells and a group of genes, including *CAND1*, *RRM2*, and *SLC2A1*, and they did not assess the correlation between the infiltration of immune cells and *CAND1* alone.

Given that *CAND1* correlates positively with DMSF and lymph nodes in ERα-positive breast cancer patients, *CAND1* might be involved in mediating invasion and metastasis in ERα-positive breast cancer patients. A published report showed similar results, in which higher *CAND1* was observed in breast cancer with lymph node metastasis compared to breast cancer without lymph node metastasis [40]. Although the role of CAND1 in mediating invasion in breast cancer has not been investigated yet, CAND1 has been shown to stabilize and synergize with PLK4 to induce centriole overduplication in prostate cancer [14]. PLK4 has been implicated in mediating metastasis and invasion by mediating the events of actin cytoskeleton [41]. These observations suggest that *CAND1* might mediate invasion and metastasis in ERα-positive breast cancer, possibly through the stabilization of PLK4. In addition, CAND1 might mediate metastasis through the interaction with miR-148b-3p, which has been implicated in mediating the migration of breast cancer cell lines [20,21]; however, the exact mechanism remains to be unidentified.

Our results showed that *CAND1* might be involved in the activation of the androgen and estrogen signaling pathways in ERα-positive breast cancer, as the genes that strongly correlate with *CAND1* in ERα-positive breast cancer were involved in activating these pathways. The androgen and estrogen signaling pathways have been shown to mediate invasion and metastasis in different cancers, including breast cancer [42,43,44,45,46]. This suggests that the correlation observed between *CAND1* and metastasis in ERα-positive breast cancer might be due to the activation of the estrogen and androgen signaling pathways. Although the role of *CAND1* in regulating the estrogen signaling pathway has not been extensively studied, it has been observed that the estrogen receptor positively regulates *CAND1* in ERα-positive breast cancer cell lines [47]. This suggests that *CAND1* might be involved in this signaling pathway. The role of *CAND1* in regulating the androgen receptor has been studied in prostate cancer, in which *CAND1* has been shown to mediate cancer aggressiveness, possibly not through the androgen receptor signaling pathway, as it was shown to mediate cancer aggressiveness in androgen receptor-positive and -negative prostate cancer cell lines [15]. Given that miR-148a regulates the expression of *CAND1* and that miR-148a is an androgen-responsive miRNA [13,48,49], the androgen signaling pathway might indirectly regulate *CAND1* or *CAND1* might play a role in this pathway. However, further experiments should be carried out to explore the involvement of *CAND1* in the androgen and estrogen signaling pathways.

Given that the results indicate that *CAND1* correlates with worse prognosis in ERα-positive breast cancer and that significantly higher *CAND1* occurs in ERα-positive breast cancer patients who received hormone therapy compared to ERα-positive breast cancer patients who did not receive hormone therapy, clinicians should be aware when using hormone therapy to treat ERα-positive breast cancer patients, as it might increase *CAND1* expression in patients.

Despite the above results and advantages of such bioinformatics analysis, some limitations need to be addressed. Specifically, the possible mechanism by which *CAND1* activates the androgen and estrogen signaling pathways remains unknown. Further mechanistic analysis of *CAND1* using in vitro and animal models is needed in order to robustly elucidate these mechanisms. Such analysis will aid in the development of novel therapeutics specifically targeted at Erα-positive breast cancer patients with upregulated *CAND1*. Moreover, combining bioinformatics with computer-aided diagnosis techniques, including cloud-based extreme learning machine, intuitionistic fuzzy active contour method, and neutosophic set-based medical image analysis, is crucial to validate our observations and to further explore the prognostic value of CAND1 in Erα-positive breast cancer, as these techniques outperform other techniques in diagnosing cancers and predicting outcomes [50,51,52].

## 5. Conclusions

In conclusion, it was shown that *CAND1* is significantly upregulated in breast cancer patients compared to normal breast tissue, and higher *CAND1* expression was observed in histologic grade 3 compared to grades 2 and 1 in ERα-positive breast cancer patients. In addition, it was found that *CAND1* correlates positively with DMSF and lymph nodes in ERα-positive breast cancer patients. These results suggest that *CAND1* could be used as a potential novel biomarker for worse prognosis in ERα-positive breast cancer, as it correlates positively with worse OS, RFS, and DFMS. Our results suggest that *CAND1* might mediate invasion and metastasis in ERα-positive breast cancer, possibly through the activation of the estrogen and androgen signaling pathways. Despite the promising results of such bioinformatics analyses, using in vitro and animal models is necessary in order to robustly elucidate the significance of *CAND1* in mediating invasion and metastasis in ERα-positive breast cancer and to explore the role of *CAND1* in activating the androgen and estrogen signaling pathways.

## Figures and Tables

**Figure 1 diagnostics-12-02327-f001:**
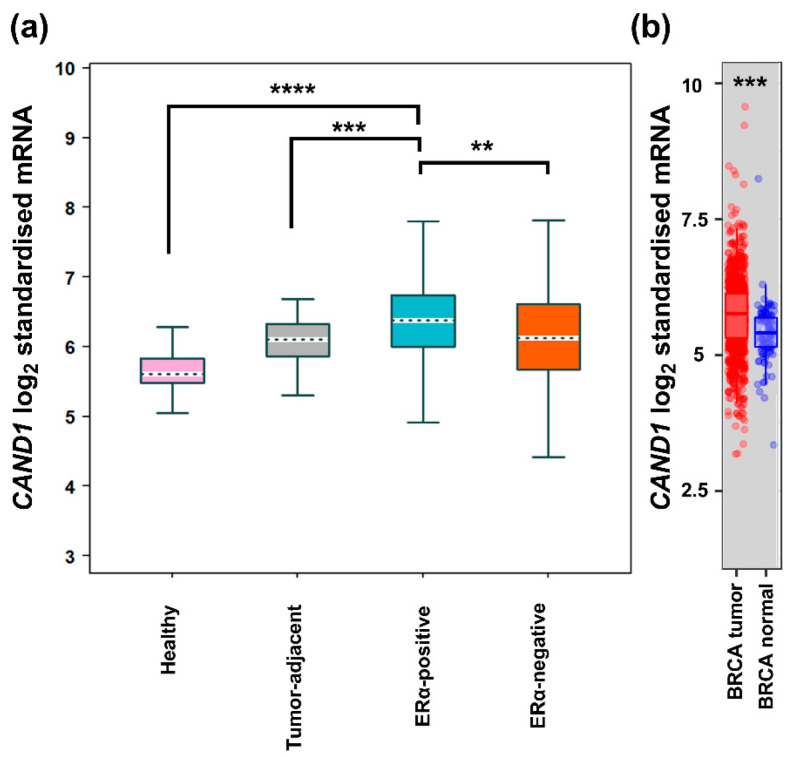
(**a**) *CAND1* mRNA expression in normal, tumor-adjacent, ERα-positive and -negative breast cancer tissues using bc-GenExMiner; (**b**) *CAND1* mRNA expression in normal and breast cancer tissue using TIMER2.0. ** *p* < 0.01, *** *p* < 0.001, and **** *p* < 0.0001.

**Figure 2 diagnostics-12-02327-f002:**
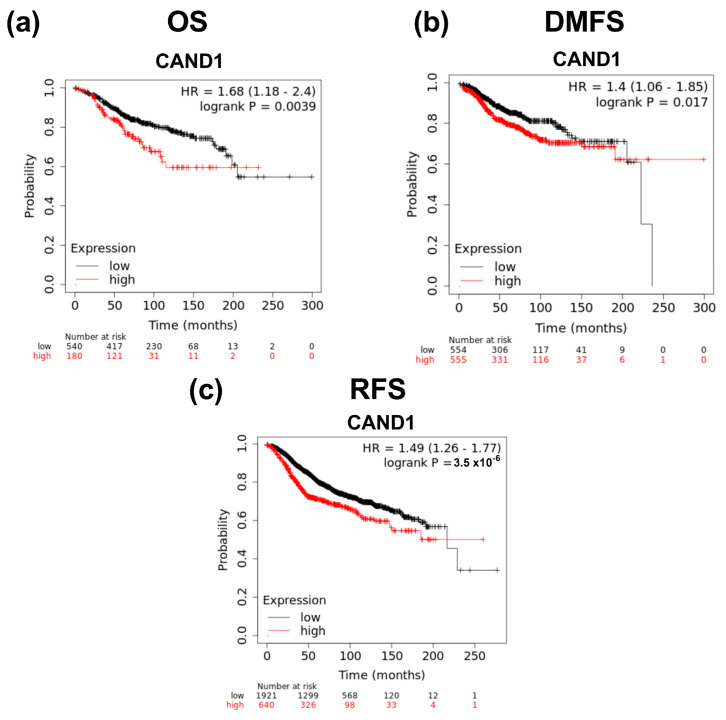
Survival curves evaluating the prognostic value of *CAND1* in ERα-positive breast cancer patients: (**a**) Analysis for OS; (**b**) analysis for DMFS; (**c**) analysis for RFS.

**Figure 3 diagnostics-12-02327-f003:**
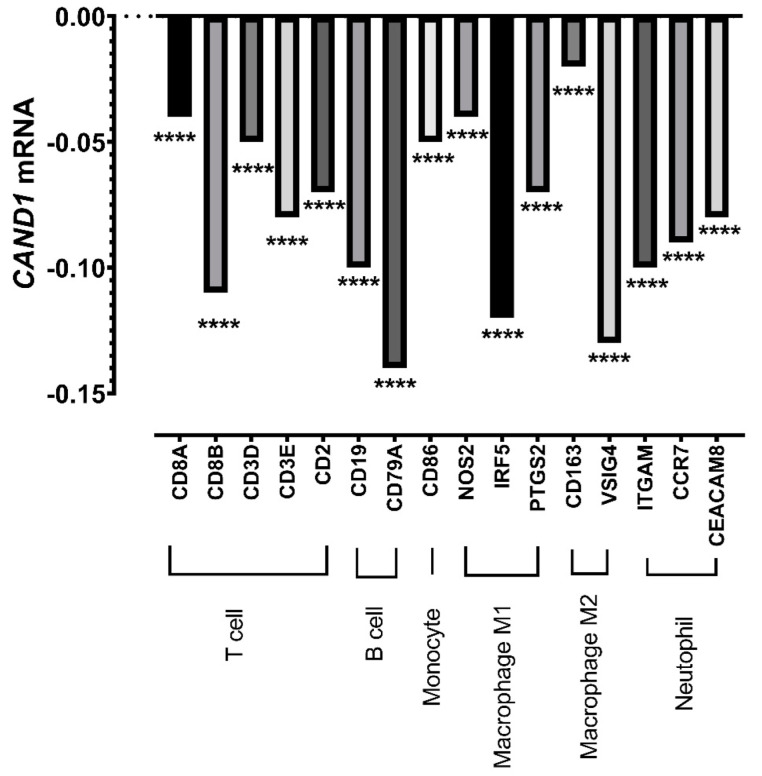
Pearson’s correlation coefficient between *CAND1* mRNA and several gene markers of immune cells in ERα-positive breast cancer patients, including T cells, B cells, monocytes, macrophages M1 and M2, and neutrophils. **** *p* < 0.0001.

**Figure 4 diagnostics-12-02327-f004:**
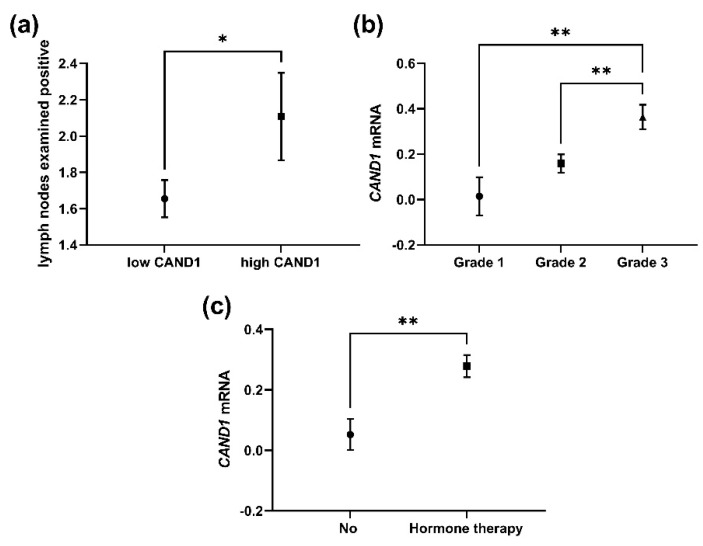
Plots evaluating *CAND1* expression in ERα-positive breast cancer patients according to different clinical parameters: (**a**) Analysis for lymph nodes examined positive; (**b**) analysis for neoplasm histologic grades; (**c**) analysis for ERα-positive breast cancer patients who received hormone therapy and patients who did not receive hormone therapy. * *p* < 0.05 and ** *p* < 0.01.

**Figure 5 diagnostics-12-02327-f005:**
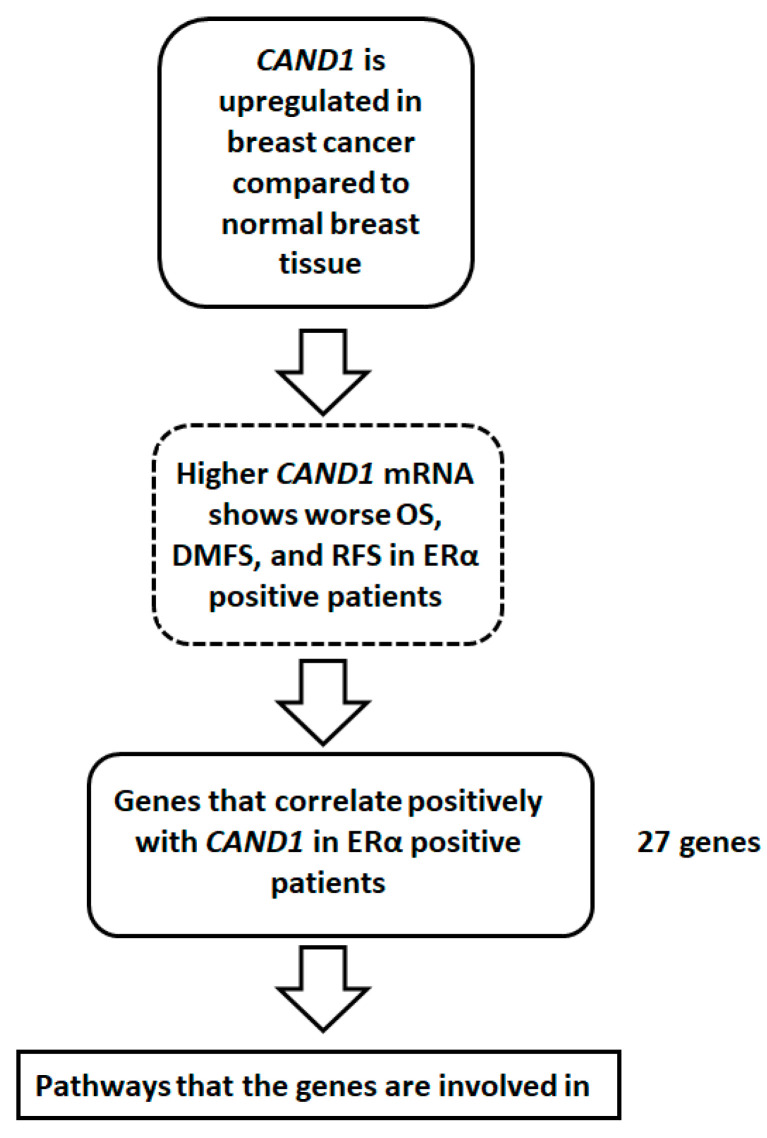
Schematic presentation of the experimental design followed in this study.

**Figure 6 diagnostics-12-02327-f006:**
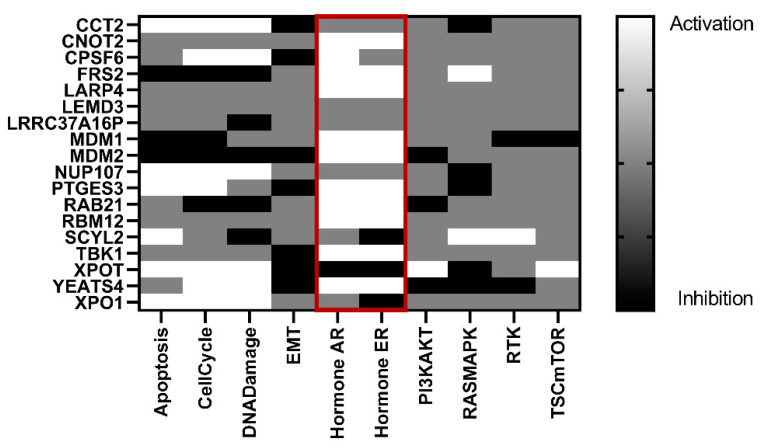
Heat map showing the activated and inhibited pathways based on genes that significantly correlate with *CAND1* in ERα-positive breast cancer patients.

**Table 1 diagnostics-12-02327-t001:** Genes that correlate positively with *CAND1* in ERα-positive breast cancer patients.

Gene Symbol	Correlation Coefficient ^1^	*p*-Value	Number of Patients
CCT2	0.6541	<0.0001	3685
LEMD3	0.6339	<0.0001	3685
NUP107	0.6324	<0.0001	3685
XPOT	0.6155	<0.0001	3685
TBK1	0.6058	<0.0001	3685
PCBP2P2	0.5962	<0.0001	530
CNOT2	0.5901	<0.0001	3685
ANKRD18EP	0.5863	<0.0001	530
XPOTP1	0.5848	<0.0001	530
YEATS4	0.5838	<0.0001	3685
CPSF6	0.5686	<0.0001	3685
RAB21	0.5648	<0.0001	3685
FRS2	0.5577	<0.0001	3685
LARP4	0.5474	<0.0001	3685
YWHAZP4	0.5464	<0.0001	530
MDM2	0.5444	<0.0001	3685
MDM1	0.5387	<0.0001	3685
SUDS3P1	0.5379	<0.0001	530
LRRC37A16P	0.5342	<0.0001	530
PTGES3P3	0.5318	<0.0001	530
SCYL2	0.5234	<0.0001	3685
PTGES3	0.5193	<0.0001	3685
NUS1P1	0.5097	<0.0001	530
FAM3C2P	0.5081	<0.0001	530
XPO1	0.5045	<0.0001	3685
TCAF1P1	0.5041	<0.0001	530
RBM12	0.5014	<0.0001	3685

^1^ Pearson’s correlation coefficient.

## Data Availability

Not applicable.

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
