# Peer review of "Bioinformatics Analysis of the Prognostic Significance of CAND1 in ERα-Positive Breast Cancer"

_diagnostics, 2022, doi:10.3390/diagnostics12102327_

Round 1

Reviewer 1 Report

Dear author, the article is well written and is focused on the up-to-date genetic background of breast cancer. I have one remark : the authors present a table of gene-gene interractions....I recommend to ad also epigenetic regulation (e.g. miRNAs) as it was published in articles like:  Kudela E, Samec M, Koklesova L, Liskova A, Kubatka P, Kozubik E, Rokos T, Pribulova T, Gabonova E, Smolar M, Biringer K. miRNA Expression Profiles in Luminal A Breast Cancer-Implications in Biology, Prognosis, and Prediction of Response to Hormonal Treatment. Int J Mol Sci. 2020 Oct 17;21(20):7691. doi: 10.3390/ijms21207691. PMID: 33080858; PMCID: PMC7589921.

Round 2

Reviewer 2 Report

Authors need to revise the paper thoroughly for English grammar bugs and phraseology carefully. Check all the abbreviation carefully and abbreviate all during its first use. Authors have not cite the most of the suggested references that matches the scope of the paper but cited number of unrelated references in the revised version of the paper. Rectify it carefully.
